# Socioeconomic status and the likelihood of informal care provision in Japan: An analysis considering survival probability of care recipients

**Yoko Ibuka** [1]*, **Yui Ohtsu** [2]

**1** Faculty of Economics, Keio University, Tokyo, Japan, **2** Graduate School of Humanities and Social Sciences, Saitama University, Saitama, Japan

* ibuka@econ.keio.ac.jp

**Data Availability Statement:** The JSTAR data are third party data, and the authors are not able to share the dataset because for data use, they have

## Abstract

Studies show that the burden of caregiving tends to fall on individuals of low socioeconomic status (SES); however, the association between SES and the *likelihood* of caregiving has not yet been established. We studied the relationship between SES and the likelihood of adults providing long-term care for their parents in Japan, where compulsory public long-term insurance has been implemented. We used the following six comprehensive measures of SES for the analysis: income, financial assets, expenditure, living conditions, housing conditions, and education. We found that for some SES measures the probability of care provision for parents was greater in higher SES categories than in the lowest category, although the results were not systematically related to the order of SES categories or consistent across SES measures. The results did not change even after the difference in the probability of parents' survival according to SES was considered. Overall, we did not find evidence that individuals with lower SES were more likely to provide care to parents than higher-SES individuals. Although a negative association between SES and care burden has been repeatedly reported in terms of care intensity, the caregiving decision could be different in relation to SES. Further research is necessary to generalize the results.

## Introduction

Health policy researchers have taken considerable interest in the influence that socioeconomic status (SES) has on individual health. Studies have repeatedly indicated a positive association between SES and a broad spectrum of health outcomes. Recent studies have extended the analysis to long-term care, as the SES gradient in long-term care, including both care receipt and care provision, is an important consideration when designing a public long-term care system, in which equality is valued. Moreover, as in many European countries, several policy reforms have recently been implemented in Japan's public long-term care system, mainly due to the state's limited fiscal capacity. These policy reforms include adjustments in eligibility rules and an increase in the coinsurance rate for long-term care services. In addition, supply-side

signed the terms and conditions with the Research Institute of Economy, Trade and Industry, which provided the JSTAR dataset. The terms and conditions include the non-transfer of data to any other person/party. Any researcher can access these datasets upon application, and the authors did not have any special access privileges that others would not have. The details of the application process is found in the following URL: https://www.rieti.go.jp/en/projects/jstar/.

**Funding:** This research is financially supported by the Open Research Areas for Social Sciences (ORA program) of the Japan Society for the Promotion of Science (JPJSJRP20181403). The funder had no role in study design, data collection and analysis, decision to publish, or preparation of the manuscript.

**Competing interests:** The authors have declared that no competing interests exist.

policies have been implemented, constraining the number of long-term care facilities covered by public insurance. This could lead to a rising demand for informal care in the future; therefore, it is necessary to understand the factors motivating the decision to provide informal care.

Studies on the relationship between SES and long-term care provision mainly focus on the association between SES and caregiving intensity, frequency, and/or duration [1–3]. These studies generally show a higher burden of care for caregivers with lower SES. However, the association between SES and an individual's decision to provide informal care is not very clear, whereas the factors motivating people to provide informal care have attracted attention [4]. On this basis, this study aims to analyze the association between SES and the likelihood of individuals providing care for parents and the underlying mechanism, using the Japanese Study of Aging and Retirement (JSTAR) datasets.

When we study the relationship between SES and the decision to participate in informal care provision, one important aspect that needs to be considered is the relationship between SES and the survival probability of care recipients. A study showed that due to selective mortality health inequality at the population level converges with increasing age [5]. In addition, the SES gradient of life expectancy has been repeatedly confirmed in many countries [6–8], with one U.S. study as an exception [9]. The observed SES gradient of life expectancy is likely to affect the relationship between SES and the probability of informal care provision, as care is provided only for those who are alive. Hence, we considered the selection of survival in our analyses to explore the relationship between SES and the provision of long-term care. We did so by weighting the sample using the inverse probability of the predicted survival of the parents [10].

In this study, we focused on adult children who provided care for their parents. Generally, spouses are the most typical source of informal care provision [11]. At the same time, adult children frequently play an essential role in caregiving. As of 2016, either co-residing adult children or non-co-residing relatives, which presumably includes the children of care recipients, accounted for one-third of elderly care cases in Japan [12]. In addition, studies emphasize the importance of the role of adult children as caregivers for parents in the Japanese context with the cultural tradition of eldest sons and their spouses taking care of elderly parents [13].

As noted in the literature [14, 15], the decision to provide informal care is a complex process, and this study does not aim to describe the complete set of determinants or all dimensions of the decision-making process within families. Rather, this study aims to provide evidence of how the propensity for care provision differs according to socioeconomic status and attempts to understand the factors in this relationship. We limited our analysis to care at home for the study of the relationship between SES and informal care provision, as informal care provision in a residential setting presents more extensive options regarding the type of care provided.

## Literature

### Determinants of care provision

Several studies have investigated the determinants of the supply of long-term care within a family. Employment or previous employment is an important factor. A study found that employment reduces willingness to provide informal care [16]. Similarly, employment status has a significant deterrent effect on care provision, with different consequences according to the type of previous work, type of care, and gender [17]. Studies also report the opposite causal direction, whereby care provision affects employment [13, 18].

Another important aspect to consider regarding the determinants of care provision is access to formal care, in which the costs and quality of care are significant issues [13]. Relatedly,

studies have attempted to estimate the monetary value of care [19, 20]. The difference in care provision by race or ethnicity is partly explained by the price differences experienced by each group [21]. A Japanese study found that the capacity of long-term care facilities has no effect on employment at the regional level, suggesting that access to facility care is less likely to be related to a balance between care provision and employment [22]. Access to formal care depends on the coverage benefits provided by public long-term care insurance (LTCI). The introduction of public LTCI has a positive impact on the employment of caregivers [23].

Some studies have assessed the heterogeneity in care provision by gender. The literature often emphasizes the role of women as caregivers. The difference in the gender proportion of those who report providing care depends on the age of the caregiver [24]. Men are more likely to share care responsibility with other informal caregivers in caring for spouses than are women [4].

## SES and care burden

Some studies have focused on the association between SES and care burden among caregivers. These studies measure care burden by considering care time [1, 3], frequency [1, 15], duration, and the degree of dependence [1]. They generally find an SES gradient of care provision, indicating that individuals with lower SES engage more intensively in informal care than individuals with higher SES. Specifically, Saito et al. report, using Japanese data, that caregivers in the lowest 25% of the income quantile are 1.43 and 1.79 times more likely to provide care for 36 and 72 hours per week, respectively, compared to caregivers in the top 25% [3]. A clinical review showed that the risk factors associated with the burden of caregiving included low levels of educational attainment [2].

Two studies indicate that inequality exists in access to formal care based on individuals' SES, suggesting a potential mechanism for explaining the relationship between SES and care burden [25, 26]. Another explanation is the difference in labor participation across SES groups. The introduction of public LTCI increased the labor participation rate of family caregivers among high-income households, possibly due to the greater opportunity costs of time [27]. Studies reveal that inequality in informal care provision varies across countries, with greater inequality in countries with lower public financing for long-term care [28].

Two studies were the closest to our study in terms of motivation. The first study used the two-part model to explore the extensive (likelihood) and intensive (magnitude) margins of informal care provision [29]. Another study reported that less-educated single women tended to be primary caregivers for parents with high disability levels [26]. We extend the analysis of the probability of care provision for both male and female caregivers using various SES measures.

## Institutional background

Compulsory public LTCI was introduced in Japan in 2000 to support the lives of the elderly. The beneficiaries are individuals aged 40 or above, and the LTCI finances use both tax (50%) and insurance premiums from beneficiaries (50%). Services covered by the LTCI include care both at home and at a facility.

Benefits are provided only in kind. To receive a service, beneficiaries apply for an assessment to determine whether they need care and how much if they do. The assessment consisted of two steps: computer assessment and interviews by experts. Based on the assessment, individuals are categorized into eight groups from "independent" to "highest-level care requirement" according to their care requirements. At the time of our analysis, individuals pay 10% of coinsurance when they receive LTC services, and those who are on public assistance are free of

charge. In addition, depending on the care requirement category, a different monthly cap is set for the total amount of home-care services, by which those with highest-level care requirements are allowed to use the most. Individuals could receive more care than the cap if they paid 100% of the charge. Services at home include helping in daily life, such as assistance in shopping, cleaning, bathing, and toileting. Services also cover nurse visits for health care and rehabilitation at home. Those who receive care at home could also visit a facility to receive day-care or short stay services.

Three major types of facilities are covered by the term "LTCI": "LTC welfare facilities" for those who are going to stay at nursing homes for their lifetime, "LTC rehabilitation" for those who are supposed to be back home after a certain period, and "LTC medical facilities" for those who require medical care. In addition, there are private nursing homes available.

## Data and methods

### Data

In this study, we used data from the Japanese Study of Aging and Retirement (JSTAR), conducted by the Research Institute of Economy, Trade and Industry (RIETI), Hitotsubashi University, and the University of Tokyo [30, 31]. The JSTAR has a consistent set of validated questionnaires with the Health and Retirement Study in the United States; the Survey of Health, Ageing and Retirement in Europe; and the English Longitudinal Study of Aging in the UK so that we can pool the data or compare the results across surveys. The response rate was 60% [31]. We used data provided by RIETI for research purposes on an application basis. The survey included a comprehensive set of questions about an individual's economic, social, and health conditions for those aged 50 to 75 years at baseline in Japan. In 2007, the survey covered five municipalities, and in 2009, two additional municipalities were included. In 2011, three cities were added to the survey's coverage, resulting in a total of 10 municipalities. In each municipality, participants were randomly chosen based on household registration [31]. The JSTAR has a longitudinal structure in which individuals can be traced over a maximum of four time points. However, our analysis is based on the first wave for each municipality conducted in 2007, 2009, or 2011; thus, our dataset is cross-sectional. This is because the purpose of this study is to describe the association between SES and informal care provision, rather than to investigate the causal impact of one on the other. In addition, we observed few changes in SES and care provision over time, producing less meaningful results from a longitudinal analysis. Our dataset consists of 7,105 individuals who responded to the first wave of the JSTAR conducted in 2007, 2009, and 2011 where at least one SES measure, gender and age were observed [30, 31].

### Methods

We conducted four analyses based on different samples. In each analysis, we used the maximum number of observations with no missing values for both the dependent and independent variables. First, to understand the association between SES and informal care provision for parents, we ran a logistic regression where the dependent variable is a binary variable that takes unity if a respondent cares for at least one parent and/or parent-in-law. The main independent variable is an SES measure, and we control for a set of respondent characteristics, as described in the Variables section. In this analysis, we restrict our sample to respondents of whom at least one parents is alive. This serves as the baseline analysis, in which we do not consider the survival probability of parents. Second, to test if there is an SES gradient in the survival of parents, we examined the relationship between SES and the selection of the target population. If we observed the SES gradient, we needed to take the gradient into consideration

as it would affect the observed relationship between SES and the probability of care provision. Specifically, we tested the hypothesis that parents of those in lower SES groups are less likely to be alive at the time of the survey, after controlling for the age and other characteristics of respondents. We used all the respondents in this analysis. Third, to consider the difference in the probability of survival according to SES, we attempted to match SES distribution using the inverse probability weighting method between respondents whose parents were alive and those whose parents were deceased. The probability weighting method is often used to deal with biases shown by non-responses in a survey [10], and is applied to non-random attrition caused by health conditions in longitudinal studies [32]. We applied the method to handle non-responses regarding care provided to parents due to their demise. The process included two steps: in the first step, we obtained the probability of parents' survival using a probit model, where the dependent variable is a binary that indicates whether the respondent's parents are alive, and computed the predicted probability for each respondent. In the second step, we regressed the care provision binary on SES and a set of respondent characteristics using the inverse of the predicted probability obtained in the first step as a weight. By doing so, we provided a greater weight, in general, to low-SES individuals compared to the baseline regression conducted in the first analysis. We used all the individuals to calculate the weight in the first step, and the second step was based on respondents whose parents were alive. Fourth, we ran two additional logistic regressions to understand how SES is (un)related to informal care provision. We considered the following two potential mechanisms to explain the relationship between SES and provision of care to parents. The first mechanism is the care requirement of parents, and we thus examined whether SES is associated with the care requirement of parents. Therefore, the first logistic regression with a binary variable to indicate the care requirements of parents was performed to examine if a difference exists in their care needs according to SES. This analysis was based on respondents whose parents were alive. The second factor is a choice between home care and residential care. We ran the second logistic regression with a binary variable that takes unity if a parent requires care at home and zero if he/she receives care from a facility. This study examined how the choice between home care and institutional care differed across SES groups. This analysis was based on respondents whose parents required care.

We included one SES measure in one regression and ran a total of six regressions in each analysis. We used 5% as the threshold for significance. For statistical analysis, we used Stata version 16.1.

## Variables

**Dependent variables.**   In the main analysis, we evaluated the association between SES and informal care provision for parents; thus, our dependent variable is a binary variable that indicates whether a respondent cares for a parent. The variable takes unity if a respondent cares for at least one parent (either his or her own parents or a spouse's parents). Unfortunately, the survey did not include questions about whether a respondent cares for parents. Thus, we produced a binary variable using the following two questions. The first question asked whether a respondent cares for family members or relatives. The second question queried the conditions of the respondent's parents and parents-in-law (if a respondent was married) whether they required care. We then produced the care provision variable as follows. First, we produced a binary that takes unity if a respondent cares for family members or relatives, and his/her father receives home-based care, and zero otherwise. Second, we repeated the first step for the respondent's mother, mother-in-law, and father-in-law, and produced at most four binary variables for each respondent. Third, we produced the final binary variable, which takes unity if at

least one of the four variables takes unity, and zero if all four variables are zero. In the analysis, we used a narrow definition of care that is associated with physical care, although informal care may involve assistance with household chores and other daily activities [33].

In the analysis of survival, we used a binary to indicate unity if the parent was alive at the time of the survey, and zero otherwise. In the analysis of the mechanisms, we used the following two dependent variables. The first dependent variable indicated the need for care. The variable takes unity if the parent receives home-based or residential care, and zero if the parent does not require care. This variable was created based on the question about the status of the parent's care requirements, with three possible answers: receiving care at home, receiving care at a facility, and no need for care. If the first two options are selected, then the care needs variable takes unity, and 0 otherwise. The second dependent variable takes unity if the parent receives home-based care (first option), and zero if the parent receives residential care (second option).

**Independent variables.** The main independent variable in our analysis was the respondents' SES status. To date, many studies have used education and income as SES measures. We used the following comprehensive set of SES measures: household income, financial assets, monthly expenditure, economic conditions, housing conditions, and education. Household income and assets are calculated as the sum of the respondents' and their spouses' income and assets. Monthly expenditure refers to household expenditures. Household income, financial assets, and monthly expenditure are recorded as intervals in the dataset based on the respondents' responses, and we replaced income by the median value of the interval. In the analysis, we used four categories defined by quantiles. We also used equivalent income and expenditure for a robustness check. The standard way of obtaining equivalent income or expenditure is to divide income or expenditure by household size. Unfortunately, in the JSTAR, there was no information on the number of household members in 2007. We thus conducted a robustness check with two alternative SES measures, using data from 2009 and 2011 (S1 Fig).

Economic conditions and housing conditions were recorded by the interviewer using a five-point scale: very poor, poor, fair, good, and very good. As the responses of "very poor" were limited, we combined "very poor" and "poor" and produced four categories, where 1 indicates "very poor" and "poor," and 4 indicates "very good." For education, we prepared the following categories: 1) less than high school, 2) high school degree, 3) some college, and 4) university degree or higher.

Six measures of SES differ in the number of observations due to the different degrees of non-response. The two variables, economic and housing conditions, are complete, and the other four measures are not. In particular, household income and financial assets are variables with a significant number of missing observations. We explored the characteristics of missing observations in two ways. First, we conducted Little's test for income and financial assets, and the result shows that the missing values in the two variables are not missing completely at random ($\chi^2$ = 113.15; p<0.001). Second, given the result of the Little's test, we test whether missing observations are due to respondents' health status measured by self-rated health (S1 Table). We do not find a systematic relationship between self-rated health and non-response to household income nor financial assets.

We also added the following control variables: age, age squared, gender, marital status (married and not widowed or divorced = 1), working status (working = 1), and year dummies. Unfortunately, information on regions is not available for the security level of the dataset used for our analysis.

# Results

## SES and care provision

Those who provide care at home for any of their parents account for 7% of respondents, and the percentage is higher for own parents than for parents of spouses (Table 1). In the sample, the percentages of respondents whose father or father-in-law were alive were 12% and 11%, respectively. The corresponding figures for mothers and mothers-in-law are greater at 34% and 31%, respectively, likely reflecting longer female life expectancies on average. There are variations in the number of observations among SES measures. A significant proportion of information regarding household income and financial assets was not shared by the respondents, while information about living and housing conditions was comprehensive, since the interviewers completed this section of the questionnaire. Thus, these SES measures were used in a complementary fashion. Fifty-two percent of the respondents were female; the mean age was 63.2 years (SD 7.04), and 79% of respondents had spouses at the time of the survey. Fifty-five percent of the respondents were employed.

**Table 1. Summary statistics.**

| | | N | Mean | Std Dev | Min | Max |
|---|---|---|---|---|---|---|
| Outcome measures | | | | | | |
| | Providing informal care at home (among respondents whose parents are alive) | | | | | |
| | To any of the parents | 3,569 | 0.070 | 0.255 | 0 | 1 |
| | To own parents | 2,649 | 0.062 | 0.242 | 0 | 1 |
| | To spousal parents | 2,106 | 0.046 | 0.209 | 0 | 1 |
| Parent's status | | | | | | |
| | Father is alive | 7,105 | 0.122 | 0.328 | 0 | 1 |
| | Mother is alive | 7,105 | 0.341 | 0.474 | 0 | 1 |
| | Father of spouse is alive | 6,293 | 0.111 | 0.314 | 0 | 1 |
| | Mother of spouse is alive | 6,293 | 0.307 | 0.461 | 0 | 1 |
| SES measures | | | | | | |
| | HH income | 3,419 | 2.413 | 1.137 | 1 | 4 |
| | HH financial asset | 4,169 | 2.345 | 1.193 | 1 | 4 |
| | Monthly expenditure | 6,002 | 2.422 | 1.071 | 1 | 4 |
| | Living condition | 7,105 | 2.315 | 0.723 | 1 | 4 |
| | Housing condition | 7,105 | 2.466 | 0.847 | 1 | 4 |
| | Education | 7,070 | 2.221 | 0.993 | 1 | 4 |
| Control variables (Respondent's characteristics) | | | | | | |
| | Female | 7,105 | 0.519 | 0.500 | 0 | 1 |
| | Age | 7,105 | 63.200 | 7.037 | 50 | 77 |
| | Married | 7,100 | 0.792 | 0.406 | 0 | 1 |
| | Working | 7,067 | 0.551 | 0.497 | 0 | 1 |

Source. JSTAR datasets of 2007, 2009, 2011 surveys.

Note. HH = household. We used only the first wave. The number of observations for providing informal care at home is based on the number of individuals whose parents are alive, and the corresponding values for other variables are based on all the individuals. Variables related to spouses' parents were defined only for married respondents. HH income, HH financial assets, and monthly expenditure are categorized based on quantiles, where 1 indicates the lowest 25%. For living and housing conditions, 1, 2, 3, and 4 indicate "very poor" or "poor," "fair," "good," and "very good," respectively, based on the assessment by interviewers. For educational attainment, 1, 2, 3, and 4 indicate "less than high school," "high school graduate," "some college," and "university degree or higher," respectively. "Married" indicates married and not divorced or widowed.

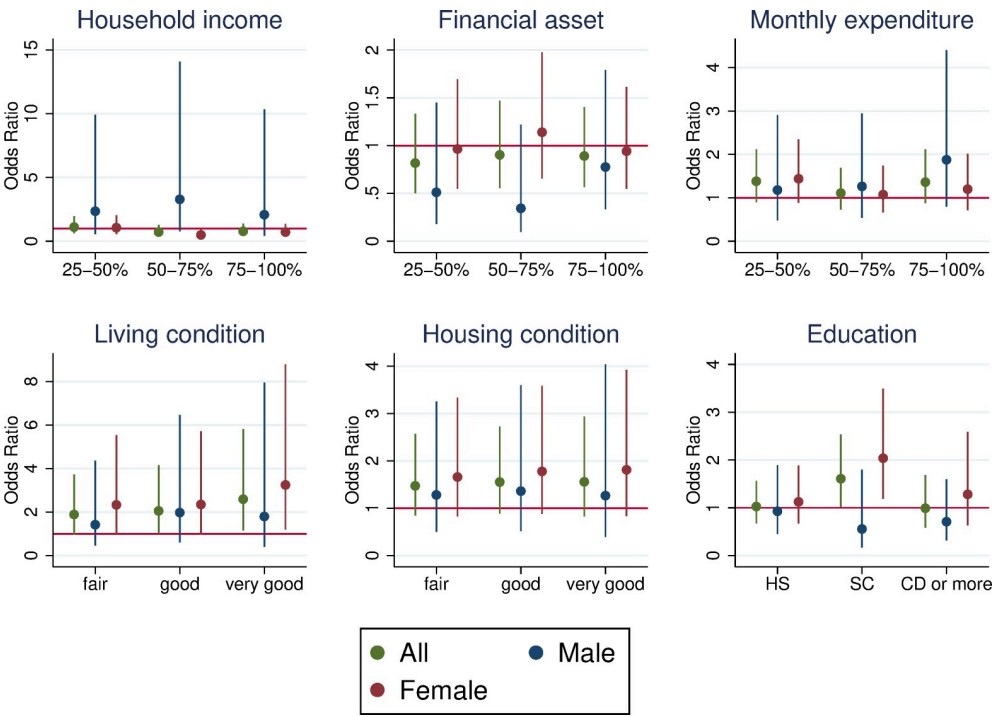

**Fig 1. Odds ratio of care provision for parents by SES.** The figure shows the odds ratio of care provision compared to the lowest SES category based on logistic regressions controlling for the age and age squared, gender (for all), marital status (married and not divorced or widowed = 1), and working status (working = 1) of the respondents. The bars represent the 95% confidence interval. The dependent variable is a binary variable that takes unity if a respondent provides care to any parent and zero if at least one parent is alive and the respondent does not provide care. Each panel shows a different SES measure. The sample of the analysis consisted of those whose parents were alive. The number of observations for all the respondents ranges from 1,708 (household income) to 3,536 (living and housing conditions).

A few significant associations between SES and care provision for parents were observed, depending on SES measures (Fig 1). The most apparent tendency was found in living conditions, and individuals with higher SES were more likely to care for parents, driven by females rather than males. Specifically, females in very good condition were 3.2 times more likely to provide care for parents than those in poor condition. A qualitatively consistent association was observed for housing conditions, although the association was not statistically significant. It was also found that females with some college education were twice more likely to provide care than those with less than high school education. In the other three SES variables, we did not find a systematic relationship between SES level and care provision. We also show the breakdown of parents or spousal parents (S2 Fig) and the same analysis where self-rated health was additionally controlled for (S3 Fig). A few positive associations between SES and care provision were found in relation to own parents. Caring for parents-in-law was very limited among males, resulting in a large standard error and imprecise estimates.

## SES and survival of parents

We found a weak positive association between SES and care provision for some SES measures in the previous analysis. One of the reasons for the positive association could be selection by attrition due to the SES gradient of survival. To investigate this possibility, we subsequently analyzed the association between the SES of respondents and the survival of parents.

Overall, we found a positive association between the survival of parents and SES of respondents (Fig 2). The association was more pronounced for mothers and mothers-in-law than for fathers and fathers-in-law. In the five measures of SES, mothers of respondents in the top-two categories were more likely to be alive than those in the lowest SES category, controlling for age of respondents and other characteristics. For mothers-in-law, a qualitatively consistent association was found. The association was less clear for fathers and fathers-in-law; however, a positive association was still found when monthly expenditure (for fathers) and living and housing conditions (for fathers-in-law) were used.

In addition, a respondent in a higher SES category systematically showed a greater odds ratio of parents' survival in many SES measures. For example, the fathers of respondents in the third and fourth quantiles of monthly expenditure were 1.4, and 1.6 times more likely to be alive, respectively, than the first quantile. Similarly, the mothers of respondents who attended college, and obtained a university or higher degree were 1.6, and 1.7 times more likely to be alive than the mothers of those with less than high school education. Given the positive association between the SES of respondents and the survival of parents, in the next subsection, we considered the selection of survival in the analysis of care provision.

## SES and care provision considering selection

The analysis in the previous subsection suggests that the higher provision of care for parents among individuals with higher SES could be explained by a higher probability of parents surviving. Hence, we assessed the association by considering the selection by survival by weighting with the inverse probability weighting. After the inverse probability weighting was applied, the mean SES of respondents whose parents were alive became closer to that of those whose parents were deceased (see S2 Table).

The qualitative associations were not consistent across SES measures, and the difference from the lowest category in the predicted probability was mostly not significant (Fig 3). The only significant difference from the bottom category was in the 50–70% group among females for the analysis with household income. In addition, there was no clear relationship between SES level and the probability of providing care. Overall, we did not find a negative association between SES and caregiving, even after considering the survival probability of parents. In the analysis using equivalent household income, males in the top category had a higher probability of care provision and females in the top category had a lower probability of care provision than those in the bottom category; however, a systematic relationship was not observed (S1 Fig).

We could identify two reasons for a hypothetical higher care burden among lower-SES individuals. First, the parents of lower SES individuals were more likely to require care, and, second, they tended to receive care at home rather than at a facility. Our analysis did not find evidence of a systematic difference by SES categories in these two cases, and hence these results could explain why we did not find an SES gradient of caregiving in Japan at least partially (S3 Table, S4 Fig).

## Discussion

We assessed the relationship between care provision for parents and SES using the JSTAR dataset. To our knowledge, this is the first study on the SES gradient of long-term care provision, considering the selection of parents. Unlike studies that examined the burden of caregiving among caregivers by SES, we did not find strong evidence of a higher burden of caregiving falling on individuals with lower SES in terms of a decision on whether one provides informal care. This result did not change after the survival probability of parents was considered.

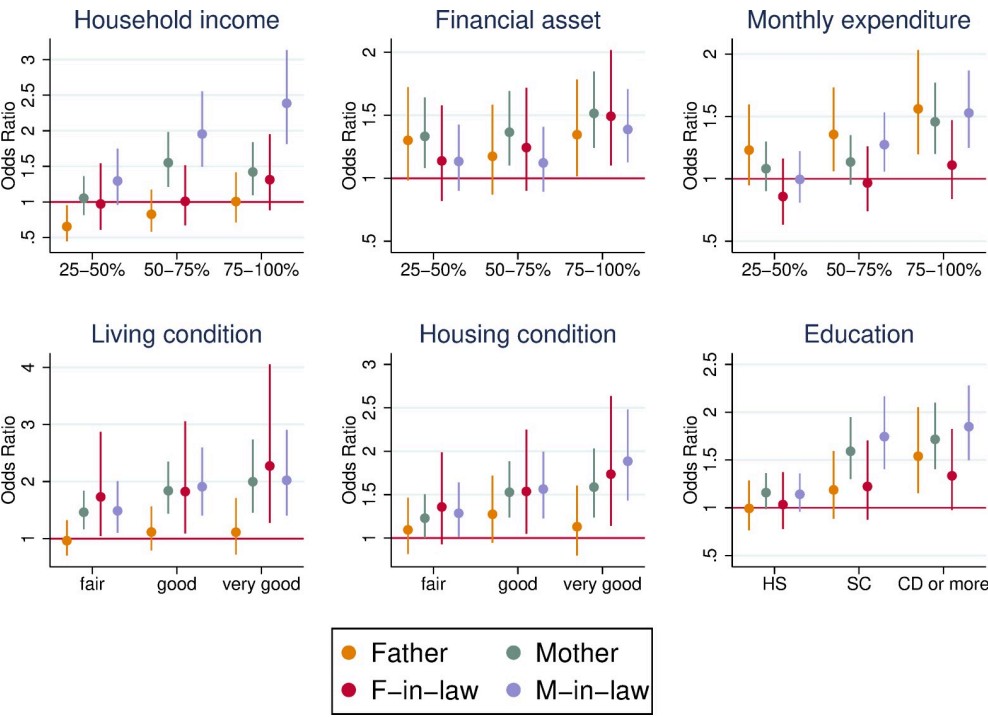

**Fig 2. Odds ratio of survival of own parents and parents-in-law by SES.** The figure shows the odds ratio of survival based on logistic regressions controlling for age and age squared, gender, and marital status (for father and mother, married and not widowed or divorced = 1) of the respondents. The bars represent the 95% confidence interval. The dependent variable is a binary variable that takes unity if the father, mother, father-in-law, or mother-in-law is alive at the time of the survey, and zero otherwise. Each panel shows a different SES measure. The sample of the analysis consisted of all respondents in the dataset. The number of observations ranges from 3,030 (parents-in-law for household income) to 7,100 (parents for living and housing conditions).

We observed a few positive associations between SES and care provision in some SES measures in the baseline analysis, which is consistent with a previous study that found that wealthier British households were more likely to provide care [29]. The relationship almost disappeared when parents' survival was considered, suggesting that the observed greater probability of caregiving among higher SES individuals was partly because their parents were more likely alive.

The finding that there was no clear association between SES and living conditions may come from the institutional background in the LTC system in Japan, where institutional long-term care services based on means testing are available for the financially disadvantaged. These facilities include moderately priced nursing for the elderly (*"Keihi roujin home"*) and care homes for the elderly (*"Yougo roujin home"*). The facilities are complementary to institutional care services covered by the public LTCI, where all the beneficiaries are eligible regardless of economics status or the availability of informal caregivers. We did not find any association between SES and living arrangements among the elderly in line with this explanation.

Studies found a greater burden of care among low-SES caregivers [1, 3] and use among low-income individuals [34]. This difference could be because our study focuses on the entry into informal care rather than care intensity. In Japan, the public LTCI was introduced in 2000, and individuals can avail themselves of the benefit of formal care services at home with 10% coinsurance. However, because of the out-of-pocket payment for receiving services, low-SES families may tend to avoid using formal care services intensively. This heterogeneity in the

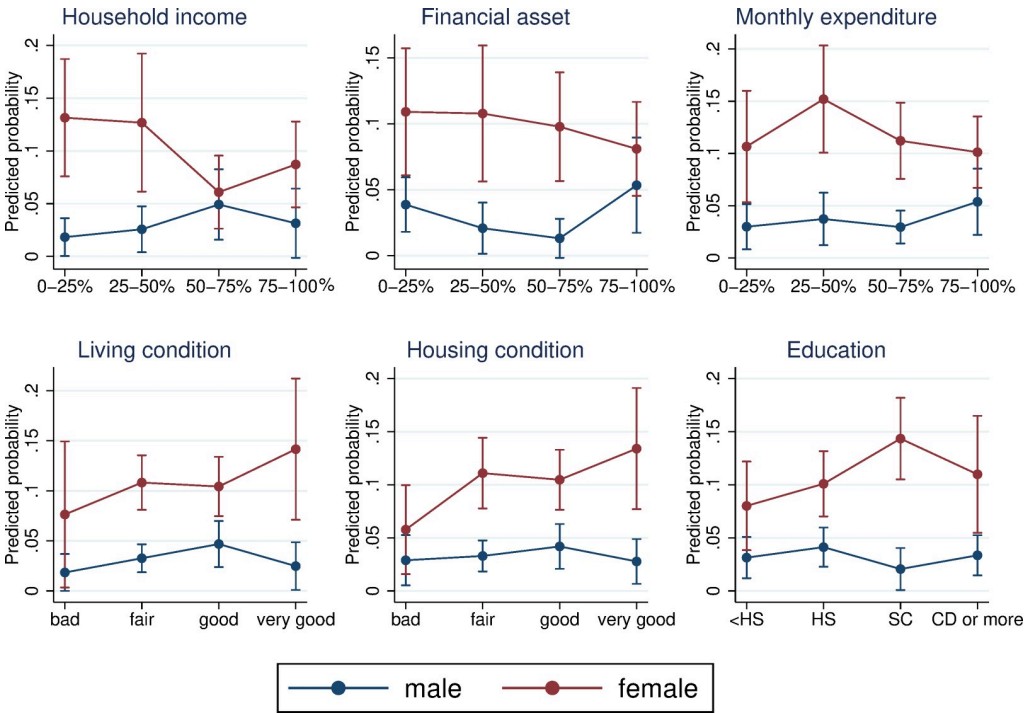

**Fig 3. Predicted probability of care provision considering heterogeneous survival probability.** The figure shows the predicted probability of care provision to parents using the inverse probability weighting. The bars represent the 95% confidence interval. In the first stage, age, age squared, gender, marital status of respondents, and year effects are included as dependent variables. The first stage was based on all the respondents. In the second stage, working status was also included, and the sample of the analysis consisted of those whose parents were alive. Each panel shows a different SES measure. The number of observations ranges from 1,710 (household income) to 3,551 (living and housing conditions).

composition of formal and informal care by SES could be a reason why the study found that low-SES caregivers tend to spend more time in providing care than high-SES caregivers. However, the heterogeneity in the composition of two types of care is unlikely to affect the propensity of caregiving, which is the target of our analysis. This is because limited research supports the claim that formal care completely replaces informal care [35], and most studies show that the formal and informal care can be provided in a complementary manner at home [36] with differing degree of disability [37], type of care [38], and country [39].

To our knowledge, no study has analyzed the association between life expectancy of parents in the context of caregiving, although our result is consistent with the findings from the following two streams of literature. First, studies have repeatedly reported a positive relationship between SES and life expectancy [6–8]. Second, numerous studies have suggested an intergenerational transfer of socioeconomic status, such as wealth [40] and education [41]. Our findings demonstrate the relationship between the life expectancy of *parents* and SES of *children*. The observed relationship between parents' life expectancy and SES of children adds a layer to the complexity of decision-making around informal care giving, in particular, with respect to the relationship with SES. Specifically, decisions on informal caregiving could differ according to caregivers' SES, as the probability of parents' being alive differs. This indicates the importance of considering the survival of parents in the analysis of SES and caregiving.

The idea of considering the selection by mortality in the analysis is similar to a study, in which non-responses due to death in a longitudinal study were adjusted using inverse

probability weighted estimators when the relationship between SES and health was assessed [32]. In this study, non-response in health status by attrition in the later waves was handled using the inverse probability weighting, whereas non-response of care-giving status due to parent's death was considered using the same method in our study.

We used six measures of SES to understand the overall view of the relationship between SES and care provision. Our results are not completely consistent across SES measures. We could offer two reasons for this inconsistency. First, for some SES measures, such as household income and financial assets, non-negligible missing observations in the dataset reduce the sample size in the analysis. If the reporting of income or financial assets by individuals is systematically related to SES status, it could affect the results and hence the difference across SES measures. Second, each SES measure can capture different aspects of informal care provision. For example, monthly expenditure could be a proxy measure of income based on the permanent income hypothesis, the economic theory that people consider not only their current income but also their lifetime income to determine their consumption, and based on the theory, monthly expenditure shows an individual's SES. At the same time, monthly expenditure could be affected by long-term care expenses, while income would not, in which case a consistent result in the analysis using income and monthly expenditure is not expected.

One of the obvious limitations is that we are not able to identify other potential informal caregivers, particularly the siblings of respondents, as the JSTAR does not request the information. If other caregivers are (primary) caregivers, it would affect the probability of care for parents, but we were not able to control for the effect due to data limitations. In addition, information on parent's attributes is unavailable in the dataset and an inclusion of variables regarding parents' characteristics could further improve the calculation of the weight. Finally, the JSTAR is not nationally representative; therefore the results of the current study are not necessarily generalizable to the Japanese population. Further research is required to investigate whether the results can be generalized using a nationally representative dataset.

## Conclusions

We assessed the relationship between care provision for parents and SES using the JSTAR dataset. Our analysis focused on the likelihood of provision of care, whereas previous studies mainly explored the differences in SES and the intensity of care provided by caregivers. Unlike previous studies that examined the burden of caregivers by SES, we did not find strong evidence that a higher burden of caregiving fell on individuals with lower SES. Although a negative association between SES and care burden has been repeatedly reported in terms of care intensity, it is important to note that the decision around caregiving could differ in relation to SES.

## Supporting information

**S1 Fig. Additional analysis using equivalent income and expenditure.** (a) Odds ratio of providing care and survival of parents; (b) care provision considering selection by survival. In the analysis in Figs 1–3, we use household income and monthly expenditure as SES measures. Although the information on household size is available only for the 2009 and 2011 surveys, we conduct the same analysis using equivalent income and expenditure, whereby household income and monthly expenditure are divided by the square root of household size. Panel (a) shows the odds ratio of providing care (top) and survival (bottom) based on logistic regressions. Panel (b) shows the predicted probability of care provision to parents using inverse probability weighting. In the first stage, age, age squared, gender, marital status of respondents, and year effects were included as dependent variables. In the second stage, the working status

was also included. In both panels, the bars represent the 95% confidence interval. Each panel shows a different SES measure. The sample used for each analysis was the same as in the main analysis. (a) The number of observations for all the respondents in the top is 900 and 1,459 for equivalent income and expenditure, and the corresponding number of observations in the bottom ranges from 1,521 (parents-in-law for equivalent income) to 2,835 (parents for equivalent expenditure). (b) The number of observations is 900 for equivalent income and 1,459 for equivalent expenditure.
(TIF)

**S2 Fig.** Care provision for parents by SES, own parents, or parents-in-law (a) Own parents (b) Spouse's parents. The figure shows the odds ratio of care provision compared to the lowest SES category based on logistic regressions controlling for the age and age squared, gender (for all), marital status (for own parents, married and not widowed or divorced = 1), working status (working = 1) of respondents. The bars represent the 95% confidence interval. The dependent variable is a binary variable that takes unity if a respondent provides care to any of the parents, and zero if at least one parent is alive and the respondent does not provide care. Estimates are missing if there are no providers in the SES category. Each panel shows a different SES measure. The sample of the analysis consisted of those whose parents were alive. (a) The number of observations for all the respondents ranges from 1,708 (household income) to 3,536 (living and housing conditions). (b) The number of observations for all the respondents ranges from 985 (household income) to 2,098 (living and housing conditions).
(TIF)

**S3 Fig. Care provision for parents by SES controlling self-rated health.** The figure shows the odds ratio of care provision compared to the lowest SES category based on logistic regressions controlling for the age and age squared, gender (for all), marital status (married and not divorced or widowed = 1), working status (working = 1), and four binary variables to show the self-rated health of the respondents. The bars represent the 95% confidence interval. The dependent variable is a binary variable that takes unity if a respondent provides care to any parent and zero if at least one parent is alive and the respondent does not provide care. Each panel shows a different SES measure. The sample of the analysis consisted of those whose parents were alive. The number of observations for all the respondents ranges from 1,708 (household income) to 3,536 (living and housing conditions).
(TIF)

**S4 Fig.** Possible mechanisms: (a) odds ratio of parents' care needs by SES; (b) odds ratio of parents receiving care at home compared to care at a facility by SES. Panel (a) shows the odds ratio of parents receiving care at home to receiving care at a facility, based on logistic regressions controlling for the age and age squared, gender, and marital status of the respondents. The coefficient shows the association between respondents' SES and the care needs of their parents when compared to the lowest SES category. The analysis was based on respondents whose parents were alive, and the number of observations ranges from 358 (household income for father-in-law) to 2,394 (living and housing conditions for mother). The bar represents the 95% confidence interval. In the first stage, the age of respondents and marital status (for the analysis of parents), as well as year effects, were included as independent variables, and in the second stage, only the age of parents and year effects are included. Each panel shows a different SES measure. Panel (b) shows the odds ratio of parents receiving care at home to receiving care at a facility among those who require care, based on logistic regressions controlling for the age and age squared, gender, and marital status of the respondents. The analysis was based on respondents whose parents required care, and the number of observations ranges from 81

(household income for father-in-law) to 680 (living and housing conditions for mother). The bar represents the 95% confidence interval. The dependent variable is a binary variable that takes unity if the father, mother, father-in-law, or mother-in-law requires care at home at the time of the survey, and zero if he/she receives care at a facility. Each panel shows a different SES measure.
(TIF)

**S1 Table. Relationship between missing observations in household income and financial assets and self-rated health.**
(PDF)

**S2 Table. Balance of SES measures with and without the inverse probability weighting.**
(PDF)

**S3 Table. Summary statistics for the outcome variables in the analyses for possible mechanisms.**
(PDF)

## Author Contributions

**Conceptualization:** Yoko Ibuka, Yui Ohtsu.

**Data curation:** Yoko Ibuka, Yui Ohtsu.

**Methodology:** Yoko Ibuka, Yui Ohtsu.

**Writing – original draft:** Yoko Ibuka.

**Writing – review & editing:** Yoko Ibuka.

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
