## [Decision Letter · Decision Letter 0]

22 Feb 2021

PONE-D-21-00550

Socioeconomic status and the likelihood of informal care provision in Japan: an analysis considering survival probability of care recipients

PLOS ONE

Dear Dr. Ibuka,

Thank you for submitting your manuscript to PLOS ONE. After careful consideration, we feel that it has merit but does not fully meet PLOS ONE’s publication criteria as it currently stands. Therefore, we invite you to submit a revised version of the manuscript that addresses the points raised during the review process.

We look forward to receiving your revised manuscript.

Kind regards,

Yu Mon Saw

Academic Editor

PLOS ONE

Journal Requirements:

2.We note that you have indicated that data from this study are available upon request. PLOS only allows data to be available upon request if there are legal or ethical restrictions on sharing data publicly. For more information on unacceptable data access restrictions, please see http://journals.plos.org/plosone/s/data-availability#loc-unacceptable-data-access-restrictions.

3.We note that the grant information you provided in the ‘Funding Information’ and ‘Financial Disclosure’ sections do not match.

4. We noted in your submission details that a portion of your manuscript may have been presented or published elsewhere.

"An earlier version of the analysis is presented at the GSA conference in November 2020 and as a result the conference abstract appears in Innovating in Aging Vol.4 with a title of “Socioeconomic status and informal care provision in Japan”.  We conduct revisions and extension of the analyses and therefore we cannot identify analyses specifically. A copy of the conference abstract is submitted as a Related Manuscript file.  "

Additional Editor comments:

We have received comments from reviewers with contradicting opinions. One reviewer recommended 'rejection,' and one suggested major revisions. Please address each comment carefully. Please also take this opportunity to improve the paper as much as possible, including ethical issues and clearly stating methodology of the study. Please ensure that the paper is aligned with the journal's guidelines and free from grammatical errors and typos. We will decide on whether to consider the manuscript further upon receiving the revised manuscript.

Reviewers' comments:

Reviewer's Responses to Questions

**Comments to the Author**

1. Is the manuscript technically sound, and do the data support the conclusions?

Reviewer #1: No

Reviewer #2: Partly

2. Has the statistical analysis been performed appropriately and rigorously? 

Reviewer #1: Yes

Reviewer #2: Yes

3. Have the authors made all data underlying the findings in their manuscript fully available?

Reviewer #1: No

Reviewer #2: Yes

4. Is the manuscript presented in an intelligible fashion and written in standard English?

Reviewer #1: Yes

Reviewer #2: Yes

5. Review Comments to the Author

Reviewer #1: The authors conducted a cross-sectional study to assess the relationship between the social economic status and informal care provision using the questionnaire survey in Japan. I agree with the importance of the topic, but this study has considerable problems mainly regarding the methodology and its consistency with the purpose of the study. Also, this manuscript partly failed to follow the PLOS ONE submission guidelines, such as the lack of ethics statement in the Methods section, which requires modification.

Major

1. Introduction, p.5, l.3-5. And Methods, p.8, l.16-p.10,l.6)

In the introduction section, the authors stated “this study aims to provide evidence of how the propensity for care provision differs by socioeconomic status and attempts to understand the factors in this relationship”, but it was not clear how this aim of this study led to the four analyses, mentioned in the Methods part (p.8).

Especially, the fourth analysis which “examined how the choice between home care and institutional care differed across SES.”(p.10, l.6), did not seem consistent with the purpose of the study, because the authors clearly mentioned that “we have limited our analysis to care at home” in the Introduction part(p.5, l.5). Please explain why all these four analyses were necessary with the view of the aim of the study.

2. Data, p.8, l.1

The authors reported 7,268 individuals as the study population, but lacked the information on i) how the authors recruited the participants (e.g., voluntarily joined, or randomly selected, etc), ii) the response rate, and iii) how the authors treated the participants with missing values. Please provide these information and discuss the generalizability of the result based on the information above.

3. Methods, p.8

Please provide the ethics statement in the Methods section. PLOS ONE submission guidelines require the authors to include ethics statements that specify the name of the approving institutional review board or equivalent committee(s) and whether informed consent was written or oral.

4. Discussion, p.20, l.1

A brief explanation of the “permanent income hypothesis” will be necessary for the readers to understand the authors’ interpretation of the result.

5. Methods, p.9, l.3. and Discussion, p.18. l.16-18.

The authors mentioned they “tested the hypothesis that parents of those in lower SES groups are less likely to be alive at the time of survey” (as their second analysis) in the Methods part, and in the Discussion part, they compared its results with other previous studies which evaluated the association between SES and life expectancy. However, what the authors evaluated was not the life expectancy since they did not seem to consider the age of parents. Please include the age of parents in the analysis, or reconsider the appropriateness of comparison of the results with other previous studies that measured life expectancy.

6. Discussion, p.19. l.3

The authors stated “Our finding adds a layer to the complexity of decision-making regarding informal care giving”, but this sentence is vague. Please explain more clearly what kind of layer this study adds, and how it should be interpreted or implemented.

Minor

1. Introduction, p.4, l.14

The authors stated “Generally, spouse tend to be the most typical source of informal care provision”, but it lacks the reference. Please add the reference.

2. Introduction, p.4, l.18

The authors argued that “studies emphasize the importance of adult children as caregivers for parents in the Japanese context”, but please explain more clearly what “the Japanese context” is.

3. Data, p.8, l.1-4.

There is no description about the basic information on JSTAR, such as where it was conducted. I suggest to add the term “in Japan”, at the end of the sentence in line 4, page 8 (…for those aged 50 years and 75 years at the baseline “in Japan”).

4. Data, p.8, l.1

Although the authors mentioned “JSTAR” stands for “Japanese Study of Aging and Retirement” in the introduction part (p.4, l.2), mentioning it again in the method part will help the readers. Please consider it.

5. Data, p.8, l.4. Table1.

The authors mentioned this dataset (JSTAR) included “those aged 50 years and 75 years at the baseline.”, but Table1 shows the youngest was 47. Please explain why it happened.

6. Data, p.8, l.2

The authors pointed “JSTAR is in harmony with the Health and Retirement Study in the US and the other surveys”, but the expressions i) “in harmony with” and ii) “other surveys” are vague. Please clarify the term i)“in harmony with”, for example, “they have the same validated questionnaire and we can compare the results”, if that is the case. Also, please specify ii)”other surveys”.

7. Methods, p.8

Please provide i)the threshold for significance, ii)the name and version of any software package used, because these information are required by the PLOS ONE Submission Guidelines.

8. Methods, p.9, l.2

The authors wrote “The main dependent variable is an SES measure”, but isn’t the SES independent variable, not the dependent variable? (as from the context of this manuscript, the dependent variable should be provision of care)

9. Independent variables, p.13, l.3-13.

This whole paragraph (The proportion of those who provide care in line 3 ~ the respondents were employed at the time of survey in line 13.) explains about the basic characteristics of the study population. I advise you to move this paragraph to the beginning of the results part.

10. Independent variables, p.13, l.3-6. Table1.

The authors described “The proportion of those who provide care at home for any of the parents is 0.07”, “the proportions of respondents whose father or father-in-law are alive are 0.12 and 0.11, respectively.”, and “The corresponding figures for mothers and mothers-in-law are by far higher at 0.33 and 0.31, respectively”. However, Table1 indicates these numbers represent mean. Is it mean or proportion? Please clarify it.

11. Independent variables, p.13, l.12-13.

The authors described “approximately half are female” and “More than half of the respondents were employed”, but please provide the number and the percentage for “approximately half” and “more than half”.

12. Independent variables, p.13, l.13.

The authors reported “the mean age is 63 years”, but please add the standard deviation when you report the mean

13. Discussion, p.19, l.14. (related to the major comment no.2)

The authors reported “there are non-negligible missing observations in the dataset that reduces the sample size in the analysis.”. If so, the author should explain how they treated missing values, and report how many study participants were excluded from the analysis.

Reviewer #2: Referee #?

PONE-D-21-00550

Comments on “Socioeconomic status and the likelihood of informal care provision in Japan: an

analysis considering survival probability of care recipients”

1. Overall evaluation

This study aims to investigate the association socioeconomic status (SES) and the likelihood of informal care provision in Japan, using the first wave of a unique longitudinal data, entitled “Japanese Study of Aging and Retirement (JSTAR)”. This study finds a new evidence that those categorized in lower SES are not necessarily more likely to provide informal long-term care (LTC) at home than those in better SES. Rather, regarding some SES measures such as self-rated living standard and female caregivers’ educational achievement, even positive correlation between SES and the probability of LTC provision were reported, after adjusting “selection bias”. The findings are not consistent to the results obtained by most previous studies.

The author(s) highlighted two principal contributions of this study; First one is focusing on the heterogeneity in the same choice set and finding that the heterogeneity would be unlikely to influence the probability of caregiving by informal/family caregivers; Second one is adjusting “selection bias” caused by parents’ death for evaluating the association of SES and LTC care provision.

I fully agree with the contributions and novelties highlighted by the author(s). Although the author(s) did not go that far in their interpretation, their findings would have a critical policy implication, such that the economic loss in the society may be more serious than we expected, because those with higher SES and higher opportunity costs are more likely to provide informal LTC.

However, I think there are several points that need to be improved in this study. I would like to provide both major and minor comments as follows. Hopefully, they are helpful for the author(s) to revise this paper.

2. Comments

2-1. Missing SES at random?

As the author(s) pointed out in Discussion (page 19), “If the reporting of income and financial assets by individuals is systematically related to SES status, it could affect the results and hence the difference across SES measures”, I doubt if income and financial assets (of which number of observations are extremely few, in particular) are missing at random. So, I suggest the author(s) to conduct a test Little’s test of missing totally at random.

Reference:

・Cheng Li. (2013). Little’s test of missing completely at random. The Stata Journal (2013)13, Number 4: 795–809.

・Little, R. J. A. (1988). A test of missing completely at random for multivariate data with missing values. Journal of the American Statistical Association 83: 1198–1202.

2-2. Health status of potential family caregivers

(1) This is also related to my first comment. I totally understand that this study focuses on the association of SES and the probability of providing informal care at home. Even so, since JSTAR surveys adults aged 50 and older, health status of potential family caregivers is a significant determinant of informal care provision, of which heterogeneity cannot be ignored. For example, health status of respondents could be mediator to associate the likelihood of answering questions with SES status. Similar to “selection” due to parents’ mortality adjusted by this study, potential family caregivers with lower health status are less likely to provide informal care to elderly persons and their lower health status could be associated with lower or missing SES.

Fortunately, JSTAR has various indicators of respondent’s health status, i.e. self-rated health status (SRH) and type of disease. Therefore, I would like to suggest the author(s) to test whether or not if a representative health status like SRH is statistically significantly correlated with the probability of missing SES.

(2) Further, regarding health status of potential family caregivers, I would like to suggest the author(s) to investigate the association of a representative indicator like SRH and the likelihood of LTC provision like SES measures. Or, the author(s) can include SRH as a control variable into regression analyses.

2-3. Heterogeneity in regions and timings of the survey

(1) This study utilizes the first wave of JSTAR datasets which surveyed 10 different municipalities and was conducted in various years, 2007, 2009, and 2011. First, the characteristics of these municipalities vary in terms of the size of population, the age structure, the fiscal scale, the geographic area, the supply of medical care and LTC, etc. I think the author(s) should include municipality dummies as covariates for adjusting these characteristics if the data are available, as these geographical characteristics might be correlated to the accessibility to formal LTC services, which would have a significant impact on the probability of demand for informal care by family members.

(2) Second, like geographical heterogeneity, the timings of the survey also vary. Although the author(s) confirmed in Data and Methods (page8), “we observed few changes in SES and care provision over time”, I still think that year dummies should include in regression analyses. For example, the global macro economy experienced Bankruptcy of Lehman Brothers in 2009 and we had the Great East Japan Earthquake in 2011, which would be influence people’s SES and thus various household-level decision makings. Further, I also suggest to include interactive terms of municipality and year dummies, because the impacts of these drastic events would vary across regions.

2-4. Japan’s LTC insurance

In order to discuss the findings, the author(s) explained the background of public LTC insurance briefly in Discussion section (p17-18). However, I think it would be helpful for international readers to understand the context of this study if the author(s) provide a brief summary for the background and system of Japan’s LTC insurance system in an additional section.

2-5. Policy implications

As I pointed out in the overall evaluation, I think this study would have a critical policy implication, such that the economic loss in the society may be more serious than we expected, because those with higher SES and higher opportunity costs are more likely to provide informal LTC. Since this study aims to describe the “correlation” between SES status and the likelihood of informal LTC provision, rather than identifying any causal relations, it is understandable why the author(s) did not go that far to interpret their findings. However, at least, the author(s) had better to mention what might happen to potential socio-economic surplus and/or social welfare if their findings are true.

2-6. Minor comments

(1) Dependent variables explained in page 11 (“the variable takes unity if the parent received home-based or residential care, and zero if the parent does not require care) are not shown in basic statistics (Table 1). The author(s) had better to show these, although they are not major dependent variables.

(2) Is the following sentence in page 12 necessarily described? “Income and expenditure could be measured at the individual level”. I think the author(s) have already explained this in the former section and this is a kind of redundant.

(3) In Table 1 (page 30), please clarify the numbers of observations (N) for outcome measures. They vary among “to any of the parents”, “to own parents”, and “to spousal parents”, because the author(s) extracted only those whose parents are alive? I might overlook the clarification in the manuscript. But, to me, it is unclear how to determine the number of samples to be regressed.

(4) Related to the above comment (3), please clarify the numbers of observations to be analyzed in all figures including the ones in Appendix. For example, are all the number of observations the same among different SES measures in Figure1 and 2? As long as I see Table 1, the number of available observations vary across these SES measures and so they must be different in Figure 1 & 2.

(5) Also, I am not clear if the author(s) run the regression including these SES variables simultaneously in the same regression estimation. Or, they put each of them in separate regression. Please specify in “Methods”.

(6) Regarding Table A1 in Appendix, please show test statistics how significantly statistically different the mean of each SES measure between two groups (parent are alive versus parents are not alive) is.

6. PLOS authors have the option to publish the peer review history of their article (what does this mean?). If published, this will include your full peer review and any attached files.

Reviewer #1: No

Reviewer #2: No

---

## [Author Response · Author response to Decision Letter 0]

17 May 2021

Response to comments from the editor and reviewers

We are grateful for the valuable comments provided by the editor and the two reviewers. We believe that incorporating these comments has helped us to improve the quality of the paper. We hereby provide our detailed responses to each comment below. 

Editor’s comments:

Thank you for the links. In the revised version, we have followed PLOS ONE’s style requirements.

2. We note that you have indicated that data from this study are available upon request. PLOS only allows data to be available upon request if there are legal or ethical restrictions on sharing data publicly. For more information on unacceptable data access restrictions, please see http://journals.plos.org/plosone/s/data-availability#loc-unacceptable-data-access-restrictions. In your revised cover letter, please address the following prompts:

We did not mean that the data was available upon request to us, and we meant that we are not able to share it with anyone. The JSTAR data we used were the secondary data, provided by the Research Institute of Economy, Trade and Industry (RIETI) of Japan, who conducted the survey in collaboration with Hitotsubashi University and the University of Tokyo. The data are owned by RIETI, and the use of the JSTAR dataset is approved by RIETI only to qualified researchers. We have signed the terms and conditions for data use, which includes the non-transfer of data to any other person/party. Therefore, under the contract with RIETI, we are unable to provide the dataset. 

The detailed rules of JSTAR data use are found at: https://www.rieti.go.jp/en/projects/jstar/. The contact information of RIETI is found below.

Research Institute of Economy, Trade and Industry (Attn: JSTAR Team)

11th floor, Annex, Ministry of Economy, Trade and Industry (METI)

1-3-1, Kasumigaseki, Chiyoda-ku

Tokyo, Japan 100-8901

b) If there are no restrictions, please upload the minimal anonymized data set necessary to replicate your study findings as either Supporting Information files or to a stable, public repository and provide us with the relevant URLs, DOIs, or accession numbers. For a list of acceptable repositories, please see http://journals.plos.org/plosone/s/data-availability#loc-recommended-repositories. We will update your Data Availability statement on your behalf to reflect the information you provide.

Please find 2-a). 

In the revised version, we have included the grant numbers for the awards for our study in the “Funding Information” section. 

4. We noted in your submission details that a portion of your manuscript may have been presented or published elsewhere. "An earlier version of the analysis is presented at the GSA conference in November 2020 and as a result the conference abstract appears in Innovating in Aging Vol.4 with a title of “Socioeconomic status and informal care provision in Japan”. We conduct revisions and extension of the analyses and therefore we cannot identify analyses specifically. A copy of the conference abstract is submitted as a Related Manuscript file. Please clarify whether this [conference proceeding or publication] was peer-reviewed and formally published. If this work was previously peer-reviewed and published, in the cover letter please provide the reason that this work does not constitute dual publication and should be included in the current manuscript. 

To apply for a presentation at the GSA meeting, we submitted an abstract with less than 250 words. There was a peer-review process in the conference application to select presenters for the conference, but no additional review process was undertaken for a publication of the abstract. The Volume 4 Supplement 1 contains all the program abstracts from the GSA 2020 annual scientific meeting (https://academic.oup.com/innovateage/issue/4/Supplement_1). The publication is a conference abstract with less than 250 words. The current paper is the full paper with more than 8,000 words; therefore, it should be considered as a distinct work from the published conference abstract. 

1. Please include captions for your Supporting Information files at the end of your manuscript, and update any in-text citations to match accordingly. Please see our Supporting Information guidelines for more information: http://journals.plos.org/plosone/s/supporting-information

We apologize for the missing information. We have included Supporting Information files at the end of the manuscript and have updated the in-text citations to correlate accordingly. 

 Additional Editor comments:

5. We have received comments from reviewers with contradicting opinions. One reviewer recommended 'rejection,' and one suggested major revisions. Please address each comment carefully. Please also take this opportunity to improve the paper as much as possible, including ethical issues and clearly stating methodology of the study. Please ensure that the paper is aligned with the journal's guidelines and free from grammatical errors and typos. We will decide on whether to consider the manuscript further upon receiving the revised manuscript.

We sincerely appreciate the opportunity to revise and resubmit the paper. We have carefully revised the paper according to comments and suggestions by the editor and reviewers. We have ensured that the paper conforms with the journal’s guidelines. 

Reviewers' comments: 

Reviewer: 1 

The authors conducted a cross-sectional study to assess the relationship between the social economic status and informal care provision using the questionnaire survey in Japan. I agree with the importance of the topic, but this study has considerable problems mainly regarding the methodology and its consistency with the purpose of the study. Also, this manuscript partly failed to follow the PLOS ONE submission guidelines, such as the lack of ethics statement in the Methods section, which requires modification.

Major

1. Introduction, p.5, l.3-5. And Methods, p.8, l.16-p.10,l.6)

In the introduction section, the authors stated “this study aims to provide evidence of how the propensity for care provision differs by socioeconomic status and attempts to understand the factors in this relationship”, but it was not clear how this aim of this study led to the four analyses, mentioned in the Methods part (p.8).

Especially, the fourth analysis which “examined how the choice between home care and institutional care differed across SES.”(p.10, l.6), did not seem consistent with the purpose of the study, because the authors clearly mentioned that “we have limited our analysis to care at home” in the Introduction part(p.5, l.5). Please explain why all these four analyses were necessary with the view of the aim of the study.

Thank you for the comments. We apologize for the insufficient clarity regarding the relationship between the objective and four analyses. Although the four analyses have different roles, they are all important in our view. We start with the analysis based on the traditional approach. Then, to show why the traditional approach is not necessarily appropriate we conduct the second analysis. The third analysis is the central analysis where we propose the application of the inverse probability weighing method for the analysis of the relationship between SES and care provision. The fourth analysis is a supplementary analysis for the mechanism to understand the main result. We believe that this supplementary analysis is particularly important as the negative association between SES and care provision has been often reported in previous studies, and we need to explain why we do not find one.

To explain our analyses better, we have extended the purpose of the analyses as follows: “On this basis, this study aims to analyze the association between SES and the likelihood of individuals providing care for parents and the underlying mechanism, using the Japanese Study of Aging and Retirement (JSTAR) datasets.” (p3, third line from the bottom). In addition, we have added the purpose of each one of the four analyses in the Methods section on pages 10 and 11.

We agree with the reviewer that the fourth analysis uses those who receive care both at home and at a facility. We have revised the manuscript as follows: “We limited our analysis to care at home for the study of the relationship between SES and informal care provision.” (p5, L5)

2. Data, p.8, l.1

The authors reported 7,268 individuals as the study population, but lacked the information on i) how the authors recruited the participants (e.g., voluntarily joined, or randomly selected, etc), ii) the response rate, and iii) how the authors treated the participants with missing values. Please provide these information and discuss the generalizability of the result based on the information above.

We apologize for the lack of information regarding the data used. The survey is not originally ours, and we use the data of Japanese Study of Aging and Retirement (JSTAR), and the dataset provided by the Research Institute of Economy, Trade and Industry (RIETI), a government research institute of Japan. Furthermore, the participants are randomly selected in the municipality of the target, and the response rate is 60%. We excluded individuals with missing values for the variables used in each analysis. We have added these explanations in the Data section (p9 – p10, L5).

3. Methods, p.8

Please provide the ethics statement in the Methods section. PLOS ONE submission guidelines require the authors to include ethics statements that specify the name of the approving institutional review board or equivalent committee(s) and whether informed consent was written or oral.

As this analysis is based on secondary data, this study is exempted from the approval of intuitional review board. 

4. Discussion, p.20, l.1

A brief explanation of the “permanent income hypothesis” will be necessary for the readers to understand the authors’ interpretation of the result.

Thank you for pointing this out. We have added an explanation of the permanent income hypothesis as follows: “the economic theory that people consider not only their current income but also lifetime income to determine their consumption” (p25, L9)

5. Methods, p.9, l.3. and Discussion, p.18. l.16-18.

The authors mentioned they “tested the hypothesis that parents of those in lower SES groups are less likely to be alive at the time of survey” (as their second analysis) in the Methods part, and in the Discussion part, they compared its results with other previous studies which evaluated the association between SES and life expectancy. However, what the authors evaluated was not the life expectancy since they did not seem to consider the age of parents. Please include the age of parents in the analysis, or reconsider the appropriateness of comparison of the results with other previous studies that measured life expectancy.

We agree that parents’ age is important for their life expectancy. In this study, the main target of the analysis is caregivers, and the purpose of the analysis regarding the survival of parents is to test if the probability of parents’ being alive would differ by SES for caregivers. To make a fair comparison across caregivers with different SES levels, we controlled for age of respondents. By doing so, we could interpret the result as the probability of parents being alive could differ by SES for a respondent at the same age. 

Additionally, the dependent variable for this analysis is a binary variable with a value of one if a parent is alive, and zero otherwise. We can observe the age of parents so long as they are alive, but we are unable to do so if the parents are dead. If we include the age of parents in the regression, we will have to give up all the data where the dependent variable takes a zero value, which makes regression analysis impossible. 

6. Discussion, p.19. l.3

The authors stated “Our finding adds a layer to the complexity of decision-making regarding informal care giving”, but this sentence is vague. Please explain more clearly what kind of layer this study adds, and how it should be interpreted or implemented.

We appreciate the comments. We have provided a comprehensive explanation as follows: 

“Our findings demonstrate the relationship between the life expectancy of parents and SES of children. The observed relationship between parents’ life expectancy and SES of children adds a layer to the complexity of decision-making around informal care giving, in particular, with respect to the relationship with SES. Specifically, decisions on informal caregiving could differ according to caregivers’ SES, as the probability of parents’ being alive differs. This indicates the importance of considering the survival of parents in the analysis of SES and caregiving.” (P24, L6)

Minor

1. Introduction, p.4, l.14

The authors stated “Generally, spouse tend to be the most typical source of informal care provision”, but it lacks the reference. Please add the reference.

Thank you. We have added the reference. 

2. Introduction, p.4, l.18

The authors argued that “studies emphasize the importance of adult children as caregivers for parents in the Japanese context”, but please explain more clearly what “the Japanese context” is.

Thank you for pointing this out. The Japanese context refers to the cultural tradition where eldest sons and their spouses tend to take care of old parents in the family. We have added the explanation. (p4, second line from the bottom) 

3. Data, p.8, l.1-4.

There is no description about the basic information on JSTAR, such as where it was conducted. I suggest to add the term “in Japan”, at the end of the sentence in line 4, page 8 (…for those aged 50 years and 75 years at the baseline “in Japan”).

We have added “in Japan.” Thank you for the suggestion. 

4. Data, p.8, l.1

Although the authors mentioned “JSTAR” stands for “Japanese Study of Aging and Retirement” in the introduction part (p.4, l.2), mentioning it again in the method part will help the readers. Please consider it.

We have written out the full name of the survey in the Methods section. Thank you for the suggestion. 

5. Data, p.8, l.4. Table1.

The authors mentioned this dataset (JSTAR) included “those aged 50 years and 75 years at the baseline.”, but Table1 shows the youngest was 47. Please explain why it happened.

Indeed, the survey includes those who are younger than 50 years old. We had four of such people in the dataset. We suspect that it is due to response error by respondents, or different persons in the household responding to the survey. Nonetheless, these people were not the target subject of the survey, and therefore, we decided to exclude them from the revised version of the analysis. Thank you for pointing this out.

6. Data, p.8, l.2

The authors pointed “JSTAR is in harmony with the Health and Retirement Study in the US and the other surveys”, but the expressions i) “in harmony with” and ii) “other surveys” are vague. Please clarify the term i)“in harmony with”, for example, “they have the same validated questionnaire and we can compare the results”, if that is the case. Also, please specify ii)”other surveys”.

Thank you. Yes, the reviewer is right that the survey has the same validated questionnaire with RHS, SHARE, and ELSA, which makes the comparison of results possible across surveys. We have explained this in the text. 

7. Methods, p.8

Please provide i)the threshold for significance, ii)the name and version of any software package used, because these information are required by the PLOS ONE Submission Guidelines.

Thank you. We have added the information in the Methods section. 

8. Methods, p.9, l.2

The authors wrote “The main dependent variable is an SES measure”, but isn’t the SES independent variable, not the dependent variable? (as from the context of this manuscript, the dependent variable should be provision of care)

This was our mistake. Thank you for pointing this out. We have revised it to “independent variable.”

9. Independent variables, p.13, l.3-13.

This whole paragraph (The proportion of those who provide care in line 3 ~ the respondents were employed at the time of survey in line 13.) explains about the basic characteristics of the study population. I advise you to move this paragraph to the beginning of the results part.

We have moved the whole sentence to the beginning of Results section, following the reviewer’s comments. 

10. Independent variables, p.13, l.3-6. Table1.

The authors described “The proportion of those who provide care at home for any of the parents is 0.07”, “the proportions of respondents whose father or father-in-law are alive are 0.12 and 0.11, respectively.”, and “The corresponding figures for mothers and mothers-in-law are by far higher at 0.33 and 0.31, respectively”. However, Table1 indicates these numbers represent mean. Is it mean or proportion? Please clarify it.

As the variables pointed out are all binary, the mean shows the proportion of those who have a unity variable. Therefore, we can interpret the mean as the proportion of the variables with a value of one. In the revised version, we have changed the numbers to those in percentage so that it would be consistent with the other descriptions. 

11. Independent variables, p.13, l.12-13.

The authors described “approximately half are female” and “More than half of the respondents were employed”, but please provide the number and the percentage for “approximately half” and “more than half”.

We have provided the specific numbers for the description.

12. Independent variables, p.13, l.13.

The authors reported “the mean age is 63 years”, but please add the standard deviation when you report the mean.

We have added the standard deviation in the text. 

13. Discussion, p.19, l.14. (related to the major comment no.2)

The authors reported “there are non-negligible missing observations in the dataset that reduces the sample size in the analysis.”. If so, the author should explain how they treated missing values, and report how many study participants were excluded from the analysis.

Thank you for providing this very helpful suggestion. The treatment of missing observations is definitely important for any analyses, and we have explained it. Our four analyses are based on different samples, and there are variations in the number of missing variables in SES. Therefore, we employ the following strategy to treat missing observations: first, we produce the largest dataset with 7,105 respondents where all the major variables are not missing (that is, at least one SES measure, gender, and age are observable). This is our original dataset. Second, in each analysis based on different subsamples from the original dataset, we use all the observations with no missing dependent and independent variable values. This way, we can take advantage of as much information as possible. We explained this in Methods section (P10, L3-9),

Reviewer’s comments:

Reviewer: 2

1. Overall evaluation 

This study aims to investigate the association socioeconomic status (SES) and the likelihood of informal care provision in Japan, using the first wave of a unique longitudinal data, entitled “Japanese Study of Aging and Retirement (JSTAR)”. This study finds a new evidence that those categorized in lower SES are not necessarily more likely to provide informal long-term care (LTC) at home than those in better SES. Rather, regarding some SES measures such as self-rated living standard and female caregivers’ educational achievement, even positive correlation between SES and the probability of LTC provision were reported, after adjusting “selection bias”. The findings are not consistent to the results obtained by most previous studies. 

The author(s) highlighted two principal contributions of this study; First one is focusing on the heterogeneity in the same choice set and finding that the heterogeneity would be unlikely to influence the probability of caregiving by informal/family caregivers; Second one is adjusting “selection bias” caused by parents’ death for evaluating the association of SES and LTC care provision. 

I fully agree with the contributions and novelties highlighted by the author(s). Although the author(s) did not go that far in their interpretation, their findings would have a critical policy implication, such that the economic loss in the society may be more serious than we expected, because those with higher SES and higher opportunity costs are more likely to provide informal LTC. 

However, I think there are several points that need to be improved in this study. I would like to provide both major and minor comments as follows. Hopefully, they are helpful for the author(s) to revise this paper. 

We sincerely appreciate the useful comments. They are very helpful to improve our paper. 

2. Comments

2-1. Missing SES at random? 

As the author(s) pointed out in Discussion (page 19), “If the reporting of income and financial assets by individuals is systematically related to SES status, it could affect the results and hence the difference across SES measures”, I doubt if income and financial assets (of which number of observations are extremely few, in particular) are missing at random. So, I suggest the author(s) to conduct a test Little’s test of missing totally at random. 

Thank you very much for the suggestion. We have conducted Little’s test for income and financial asset, and the results are reported (P15, L6) Based on Little’s test, household income and financial assets are not missing completely at random. 

2-2. Health status of potential family caregivers 

(1) This is also related to my first comment. I totally understand that this study focuses on the association of SES and the probability of providing informal care at home. Even so, since JSTAR surveys adults aged 50 and older, health status of potential family caregivers is a significant determinant of informal care provision, of which heterogeneity cannot be ignored. For example, health status of respondents could be mediator to associate the likelihood of answering questions with SES status. Similar to “selection” due to parents’ mortality adjusted by this study, potential family caregivers with lower health status are less likely to provide informal care to elderly persons and their lower health status could be associated with lower or missing SES. 

Fortunately, JSTAR has various indicators of respondent’s health status, i.e. self-rated health status (SRH) and type of disease. Therefore, I would like to suggest the author(s) to test whether or not if a representative health status like SRH is statistically significantly correlated with the probability of missing SES. 

Thank you very much for the suggestion. To determine whether missing observations in an SES measure are related to health status, we have employed a regression analysis where the dependent variable is a binary variable to indicate if an SES variable is missing and the dependent variable is self-rated health, and the result is shown in S1 Table in the Supporting Information. Our results do not show a systematic relationship between self-rated health and household income nor financial assets. 

(2) Further, regarding health status of potential family caregivers, I would like to suggest the author(s) to investigate the association of a representative indicator like SRH and the likelihood of LTC provision like SES measures. Or, the author(s) can include SRH as a control variable into regression analyses. 

We have also conducted the analysis by including series of dummy variables to show self-rated health (SRH) as control variables, and the result is shown in S3 Fig in the Supporting Information. The results did not change from those obtained from the analysis without SRH variables.

2-3. Heterogeneity in regions and timings of the survey 

(1) This study utilizes the first wave of JSTAR datasets which surveyed 10 different municipalities and was conducted in various years, 2007, 2009, and 2011. First, the characteristics of these municipalities vary in terms of the size of population, the age structure, the fiscal scale, the geographic area, the supply of medical care and LTC, etc. I think the author(s) should include municipality dummies as covariates for adjusting these characteristics if the data are available, as these geographical characteristics might be correlated to the accessibility to formal LTC services, which would have a significant impact on the probability of demand for informal care by family members. 

Thank you for the comments. We fully agree with the reviewer that it is better to control for municipality fixed effects. Unfortunately, the dataset we used for the analysis information on regions (municipalities) is masked. Moreover, the structure of our dataset allows us to include only either municipality fixed effects or year fixed effects, but not both. This is because we use a cross-sectional dataset which consists of the first waves of each municipality, and there is at a single year point data in a single municipality. Thus, in our analyses we add year dummies only, which is the best we could do given the constraint. 

(2) Second, like geographical heterogeneity, the timings of the survey also vary. Although the author(s) confirmed in Data and Methods (page8), “we observed few changes in SES and care provision over time”, I still think that year dummies should include in regression analyses. For example, the global macro economy experienced Bankruptcy of Lehman Brothers in 2009 and we had the Great East Japan Earthquake in 2011, which would be influence people’s SES and thus various household-level decision makings. Further, I also suggest to include interactive terms of municipality and year dummies, because the impacts of these drastic events would vary across regions. 

Thank you for the suggestion. In all the previous analyses, we included year dummies as control variables. For the reason stated in 2-3-(1), we are unable to include municipality dummies or interaction terms between municipality and year dummies. 

2-4. Japan’s LTC insurance 

In order to discuss the findings, the author(s) explained the background of public LTC insurance briefly in Discussion section (p17-18). However, I think it would be helpful for international readers to understand the context of this study if the author(s) provide a brief summary for the background and system of Japan’s LTC insurance system in an additional section. 

Thank you very much for the suggestion. I have now included a new section of the institutional background starting on page 7. I hope this is helpful for readers. 

2-5. Policy implications 

As I pointed out in the overall evaluation, I think this study would have a critical policy implication, such that the economic loss in the society may be more serious than we expected, because those with higher SES and higher opportunity costs are more likely to provide informal LTC. Since this study aims to describe the “correlation” between SES status and the likelihood of informal LTC provision, rather than identifying any causal relations, it is understandable why the author(s) did not go that far to interpret their findings. However, at least, the author(s) had better to mention what might happen to potential socio-economic surplus and/or social welfare if their findings are true. 

We truly appreciate the comments. We have seriously considered the comment by reviewers, and decided not to include the interpretation in Discussion section. This is because the results are mainly found among females, and SES of females and contribution to the economy are not directly related to each other in Japan. In our data, except for education, all SES variables are measured at the household level. This means, for example, a wife in higher economic conditions could not necessarily contribute to the economy more than a corresponding person in low economic conditions as she may be in a better position because of other family member who earns a lot. In addition, household wives may anyway not work regardless of caregiving status. Perhaps we could say more about males as the majority of them work, however, we did not find a positive association for males. 

2-6. Minor comments 

(1) Dependent variables explained in page 11 (“the variable takes unity if the parent received home-based or residential care, and zero if the parent does not require care) are not shown in basic statistics (Table 1). The author(s) had better to show these, although they are not major dependent variables. 

Since the analysis is independent from the main analysis, we have prepared a new table to show summary statistics in S3 Table. Notice that all the variables therein are binaries, and we show the proportion of the respondents only. 

(2) Is the following sentence in page 12 necessarily described? “Income and expenditure could be measured at the individual level”. I think the author(s) have already explained this in the former section and this is a kind of redundant. 

We have removed the sentence following the reviewer’s suggestion. 

(3) In Table 1 (page 30), please clarify the numbers of observations (N) for outcome measures. They vary among “to any of the parents”, “to own parents”, and “to spousal parents”, because the author(s) extracted only those whose parents are alive? I might overlook the clarification in the manuscript. But, to me, it is unclear how to determine the number of samples to be regressed. 

Yes, the reviewer is right. The population for the statistics is those people of whom at least one of the parents is alive. We have clarified this in the footnote of Table 1. 

(4) Related to the above comment (3), please clarify the numbers of observations to be analyzed in all figures including the ones in Appendix. For example, are all the number of observations the same among different SES measures in Figure1 and 2? As long as I see Table 1, the number of available observations vary across these SES measures and so they must be different in Figure 1 & 2. 

Thank you for pointing this out. We have now added the number of observations in each analysis in the appendices of the figures. As the number of observations depend on the measure of SES and the target of analysis, we have shown them as a range. 

(5) Also, I am not clear if the author(s) run the regression including these SES variables simultaneously in the same regression estimation. Or, they put each of them in separate regression. Please specify in “Methods”. 

We include one SES measure in one regression and run six regressions for each analysis. We have clarified this in the Methods section (P12, L10). 

(6) Regarding Table A1 in Appendix, please show test statistics how significantly statistically different the mean of each SES measure between two groups (parent are alive versus parents are not alive) is. 

Thank you for the suggestion. We have now included test statistics in S2 Table (former Table A1).

---

## [Decision Letter · Decision Letter 1]

8 Jun 2021

PONE-D-21-00550R1

Socioeconomic status and the likelihood of informal care provision in Japan: An analysis considering survival probability of care recipients

PLOS ONE

Dear Dr. Ibuka,

Thank you for submitting your manuscript to PLOS ONE. After careful consideration, we feel that it has merit but does not fully meet PLOS ONE’s publication criteria as it currently stands. Therefore, we invite you to submit a revised version of the manuscript that addresses the points raised during the review process.

We look forward to receiving your revised manuscript.

Kind regards,

Yu Mon Saw

Academic Editor

PLOS ONE

Journal Requirements:

Reviewers' comments:

Reviewer's Responses to Questions

**Comments to the Author**

1. If the authors have adequately addressed your comments raised in a previous round of review and you feel that this manuscript is now acceptable for publication, you may indicate that here to bypass the “Comments to the Author” section, enter your conflict of interest statement in the “Confidential to Editor” section, and submit your "Accept" recommendation.

Reviewer #1: (No Response)

Reviewer #2: All comments have been addressed

2. Is the manuscript technically sound, and do the data support the conclusions?

Reviewer #1: Partly

Reviewer #2: Partly

3. Has the statistical analysis been performed appropriately and rigorously? 

Reviewer #1: Yes

Reviewer #2: Yes

4. Have the authors made all data underlying the findings in their manuscript fully available?

Reviewer #1: (No Response)

Reviewer #2: Yes

5. Is the manuscript presented in an intelligible fashion and written in standard English?

Reviewer #1: Yes

Reviewer #2: Yes

6. Review Comments to the Author

Reviewer #1: Comments to the authors

The authors revised the manuscript to meet my previous comments. However, the following concern remain, which should be addressed.

1. Discussion part, p.22, l.11-12, Conclusion p.26, l.14-15

The authors mentioned “we did not find strong evidence of a higher burden of caregiving falling on individuals with lower SES.” in the discussion part.

However, the authors reached the conclusion that says “Although a negative association between SES and care burden has been repeatedly reported in terms of care intensity, it is important to note that the decision around caregiving could differ in relation to SES”.

Reviewer #2: The author(s) clarified all my questions and revised following my comments as possible as they can. However, the limitation of the data seems to be quite serious that the author(s) should clarify the further analyses to be necessary.

7. PLOS authors have the option to publish the peer review history of their article (what does this mean?). If published, this will include your full peer review and any attached files.

Reviewer #1: No

Reviewer #2: No

---

## [Author Response · Author response to Decision Letter 1]

27 Jun 2021

Response to comments from the editor and reviewers

We are grateful for the valuable comments provided by the editor and the two reviewers. We believe that incorporating these comments has helped us to improve the quality of the paper. We hereby provide our detailed responses to each comment below. We have also made minor revisions for better readability and these changes are tracked. 

Reviewers' comments: 

Reviewer 1: 

The authors revised the manuscript to meet my previous comments. However, the following concern remain, which should be addressed.

1. Discussion part, p.22, l.11-12, Conclusion p.26, l.14-15

The authors mentioned “we did not find strong evidence of a higher burden of caregiving falling on individuals with lower SES.” in the discussion part.

However, the authors reached the conclusion that says “Although a negative association between SES and care burden has been repeatedly reported in terms of care intensity, it is important to note that the decision around caregiving could differ in relation to SES”.

We apologize for the unclear explanation. We revised the first sentence as follows: 

“Unlike studies that examined the burden of caregiving among caregivers by SES, we did not find strong evidence of a higher burden of caregiving falling on individuals with lower SES in terms of a decision on whether one provides informal care.”

Reviewer 2:

The author(s) clarified all my questions and revised following my comments as possible as they can. However, the limitation of the data seems to be quite serious that the author(s) should clarify the further analyses to be necessary.

Thank you for pointing this out. We expanded the limitation according to reviewer’s comments. Also, we stated that further research is necessary in the abstract.

---

## [Decision Letter · Decision Letter 2]

2 Aug 2021

Socioeconomic status and the likelihood of informal care provision in Japan: An analysis considering survival probability of care recipients

PONE-D-21-00550R2

Dear Dr. Ibuka,

We’re pleased to inform you that your manuscript has been judged scientifically suitable for publication and will be formally accepted for publication once it meets all outstanding technical requirements.

Kind regards,

Yu Mon Saw

Academic Editor

PLOS ONE

Additional Editor Comments (optional):

Reviewers' comments:

Reviewer's Responses to Questions

**Comments to the Author**

1. If the authors have adequately addressed your comments raised in a previous round of review and you feel that this manuscript is now acceptable for publication, you may indicate that here to bypass the “Comments to the Author” section, enter your conflict of interest statement in the “Confidential to Editor” section, and submit your "Accept" recommendation.

Reviewer #1: All comments have been addressed

2. Is the manuscript technically sound, and do the data support the conclusions?

Reviewer #1: Yes

3. Has the statistical analysis been performed appropriately and rigorously? 

Reviewer #1: Yes

4. Have the authors made all data underlying the findings in their manuscript fully available?

Reviewer #1: No

5. Is the manuscript presented in an intelligible fashion and written in standard English?

Reviewer #1: Yes

6. Review Comments to the Author

Reviewer #1: They follow thatall comment

It can be published

They followed our comment properly

This is a valuable your paper

7. PLOS authors have the option to publish the peer review history of their article (what does this mean?). If published, this will include your full peer review and any attached files.

Reviewer #1: No

---

## [Editor Report · Acceptance letter]

5 Aug 2021

PONE-D-21-00550R2 

Socioeconomic status and the likelihood of informal care provision in Japan: An analysis considering survival probability of care recipients 

Dear Dr. Ibuka:

I'm pleased to inform you that your manuscript has been deemed suitable for publication in PLOS ONE. Congratulations! Your manuscript is now with our production department. 

Kind regards, 

on behalf of

Dr. Yu Mon Saw 

Academic Editor

PLOS ONE